

# Variation of the interspecific forest mass–density relationship along gradients of leaf area and global radiation

Jan J. M. van Roestel[1]

[1]Antea Group, Oosterhout, 4904 SJ, Netherlands (retired)

*Correspondence to*: Jan J. M. van Roestel (jan.vanroestel@outlook.com)

**Abstract.** Stand mass scales as the -1/3 exponent of plant density for large compilations of plant communities on a continental or global scale, being the slope of the regression line in a log–log plot, where the intercept is a normalization constant reflecting the assumption of a constant rate of energy use by the species and environments involved. Here the normalization constant is replaced by a light absorption function, enabling to investigate how the interspecific mass–density relationship varies along 10 spatial, largely latitudinal gradients of leaf area and the sum of global radiation over the growing season for relatively undisturbed forests. The test of the model for globally distributed forest communities shows the highest explained variance when both gradients are included in the light absorption function, meaning that the exponent is determined not only by the rate but also the sum of energy use over the growing season. The exponent of tree density converges to 1/2 instead of the expected 1/3 value based on the -1/3 exponent value in the bivariate biomass–density relationship. The 1/2 value corresponds with the 15 so-called self-thinning rule that applies to the self-thinning line constructed as the upper boundary of mass–density points for monospecific even-aged plant stands, where gradients in energy use can be neglected. The results demonstrate the appropriateness of introducing a light absorption function in the bivariate mass–density relationship, suggesting a thermodynamic interpretation that may be of interest to other plants and even animals when gradients in energy use similarly affect the intercept and slope of the interspecific mass–density relationship.

**1 Introduction**

For large compilations of plant communities on the continental or global scale, the allometric relationship between average living aboveground biomass per plant in an area $\bar{M}$ (g) and the plant number in that area $N$ (m⁻²) is generally well fit by the equation:

$$N_{crit} = k\bar{M}^{-3/4} \tag{1}$$

where $N_{crit}$ is the critical density of maximally packed individuals where all resources are used, with average mass $\bar{M}$ (Enquist et al., 1998; Niklas et al., 2003; Deng et al., 2012; Dillon et al., 2019). The -3/4 exponent of $\bar{M}$ is the scaling exponent and k is the scaling coefficient or normalization constant that adjusts the general relationship across environments and species. Earlier studies traditionally treated $N$ as the independent variable and $\bar{M}$ as the dependent variable, such that the exponent would be -4/3 rather than -3/4 (Weller, 1989; Lonsdale, 1990). This results in the general mass–density equation:

$$\bar{M} = kN^{-\beta'} \tag{2}$$



where the expected value of the scaling exponent -β' is -4/3 for large compilations of plant communities. For statistical reasons, Eq. 2 is preferred for forests because the number of trees in an area can be determined much more accurately as the independent variable in a linear regression model where the equation is plotted on logarithmic axes:

$$\log\overline{M} = \log k - \beta' \log N \tag{3}$$

where -β' is the slope of the line and $\log k$ the y-intercept.

Mechanistic self-thinning theories explain the -β' = -4/3 value of the scaling exponent in interspecific scaling of large compilations of plant communities from the underlying processes, using geometric, allometric and dynamic growth arguments (e.g., Weller, 1987b; Lonsdale, 1990; Adler, 1996; Enquist et al., 1998; Li et al., 2000; Deng et al., 2012), but important variation in the value of the scaling exponent and scaling coefficient remains when particular species or habitats are considered

(Deng et al., 2012).

In intraspecific scaling the thinning slope -β' (Eq. 3) converges to -3/2 when the thinning line is determined as the upper boundary of data points in a $\log\overline{M} - \log N$ plot of crowded even-aged monospecific plant populations (Yoda et al., 1963; Westoby, 1984). This so-called 'self-thinning rule' (Yoda et al., 1963; Westoby, 1984) was for some time thought also to apply to the self-thinning line obtained by tracking a community through time or by the juxtaposition of datapoints of separate stands

with a corresponding species composition (e.g., Westoby, 1984; White, 1985), until scrutiny showed a large variation in the thinning slopes and intercepts (Weller, 1987a).

So, the -4/3 exponent in interspecific scaling has predictive power on the continental or global scale, whereas the -3/2 self-thinning rule applies only to the upper boundary of datapoints in the intraspecific mass-density relationship for well-defined local conditions. Here, the explanation for this discrepancy is sought in the scaling coefficient $k$, assumed to represent a

constant high rate of energy use in the self-thinning trajectory, both for an exponent value of -4/3 in interspecific scaling and an exponent value of -3/2 according to the self-thinning rule.

The variation in the value of the scaling coefficient among thinning lines has received relatively little attention (Dillon et al., 2019), although the estimate of the variation in the intercept (i.e., $\log k$) is correlated with the estimate of the slope in the log–log relationship of Eq. 3 (Westoby, 1984). The value of the scaling coefficient is described as a measure of a constant whole

stand energy use (e.g., Westoby, 1984; White et al., 2007; Deng et al., 2012), with variations due to differences in resource use through time that are much stronger than variations in the exponent (Deng et al., 2006; Dai et al., 2009).

Light experiments on monospecific even-aged plant populations, comparing self-thinning trajectories for different but constant levels of illumination, show that the intercept of thinning lines is lowered with increasing shade, while the scaling exponent at each level of shade is maintained at a value of approximately -3/2, except for populations grown under deep shade (Hiroi and

Monsi, 1966; Lonsdale and Watkinson, 1982, 1983; Hutchings and Budd, 1981; Westoby and Howell, 1981; Westoby, 1984). This suggests that reduced light absorption due to a lower leaf area, rather than increased shade, also results in a lower intercept of the thinning line, but only a different slope if light absorption is not constant throughout the trajectory of self-thinning. The



effect of gradients in leaf area, and thus light absorption, on the exponent of $N$ in the interspecific mass–density relationship is investigated by developing a light absorption function that includes leaf area to replace the normalization constant k.

The introduction of a light absorption function also builds on the approach of Deng et al. (2012), describing the scaling coefficient $k$ in interspecific scaling as an empirically determined measure of a constant rate of total energy use, equal to $N_{crit}\bar{M}^{3/4}$ or $\bar{M}N_{crit}^{4/3}$, where the last expression considers $N$ as the independent variable, conform the 'classical' self-thinning theory (e.g., Westoby, 1984). However, this approach does not include the finding that at northern latitudes, tree density increases with decreasing latitude to approximately $25°$, while the total aboveground biomass is supposed to be constant

(Enquist and Niklas, 2001) or increases, apart from the spatially restricted temperate rainforests (Pan et al., 2013). Therefore, $\bar{M}N_{crit}^{4/3}$ increases with decreasing latitude, together with an increase in the sum of available solar energy use over the growing season, here investigated by including this gradient in a light absorption function, in addition to leaf area.

  The leaf area and the available solar radiation over the growing season are introduced stepwise in the light absorption function that replaces the scaling coefficient $k$, to correct for gradients in the total energy use of forests, which implies that not only

forests with $N_{crit}$, but more forests with tree density $N$ can be included in the test of the equations (see Methods). This results in three energy–biomass–density relationships or EBDs that enable to examine how the interspecific mass–density relationship varies along gradients of leaf area and available solar energy separately and together.

  In this introduction has been referred to the allometric mass–density relationship using $\bar{M}$, conform most cited studies, but the model development in this paper will be based on the equations written in terms of the total living aboveground biomass $M$ (g

m$^{-2}$), because the calculation of $\bar{M}$ as $M/N$ from the available field data introduces artificially inflated correlations (Weller, 1987a). The use of $M$ leads to an exponent $-\beta = -\beta' + 1$, while $k$ stays the same (see Methods). In addition to the three EBDs, the bivariate mass–density relationship and the relationship between leaf area and stand density are calculated, giving insight into how the slope of the mass–density line relates to the gradient in leaf area. The analysis focuses on the development of the exponent of $N$ because this is a central issue in the debate on the mathematical form of the self-thinning equation applied to

forests. In addition, we examine the extent to which the regression coefficients support the conclusions with respect to the development of the exponent in the EBDs.

  The EBDs assume a constant regime of light absorption over the years that is long enough to establish a dynamic equilibrium with the aboveground living biomass and tree density. Human, biotic and abiotic disturbances like thinning (not self-thinning), insect diseases and drought stress can lead to deviations in leaf area from this dynamic equilibrium, due to functional responses

of forests to disturbances (Jump et al., 2017). Therefore, only relatively undisturbed stands are included in the test of the model equations. The model equations are applied to the field data of forest biomass and density in the compendium of Cannell (1982) that comprises standardized tabulations of field and experimental data of forests of approximately 600 reports worldwide.





## 2 Methods

### 2.1 Introduction of a light absorption function in the self-thinning equation

The development of a light absorption function begins with the formulation of a balance equation underlying the self-thinning equation. The balance equation for an even-aged monospecific tree stand at the ceiling (or maximum) leaf area can be formulated by describing the scaling exponent -β' as the balance between the relative growth rate (*RGR*) and the relative mortality rate (*RMR*) of a plant stand (Hozumi, 1977):

$$\frac{RGR}{RMR} = \frac{\frac{d\overline{M}}{\overline{M}dt}}{\frac{dN}{Ndt}} = \frac{dlog\overline{M}}{dlogN} = -\beta' \tag{4}$$

This is an equation that can also be written as the next balance equation:

$$\frac{dM}{M} + \frac{\beta dN}{N} = dlogM + \beta dlogN = 0 \tag{5}$$

where $M$ is used instead of $\overline{M}$ in this equation because the use of the average aboveground body mass, calculated as $\overline{M} = M/N$, introduces artificially inflated correlations between $\overline{M}$ and $N$ (Weller, 1987a; Li et al., 2006), which impedes an adequate comparison with the other model equations. The introduction of $M$ in the equation results in an 1/2 value of the exponent β of

the tree density, when we consider the self-thinning rule, because:

$$M = \overline{M}N \propto N^{-\frac{3}{2}+1} \tag{6}$$

The integration of the balance equation (Eq. 5) results in the empirical self-thinning equation (Eq. 3A), with $logk$ as the integration constant:

$$logM = logk - \beta logN \tag{3A}$$

or more concise:

$$M = \overline{M}N = kN^{-\beta'+1} = kN^{-\beta} \tag{2A}$$

At the ceiling leaf area, the light absorption of a tree stand is maximal, and the yearly solar energy absorption regime is assumed to be in a steady state, which means that the change in yearly solar energy absorption is zero, a value that corresponds to the zero value of the right-hand term in the balance equation. Following integration of the balance equation, the light-dependent

intercept $logk$ reflects a constant level of solar energy absorption at the ceiling leaf area, which will be adjusted by the introduction of gradients in leaf area and the available solar radiation in the right-hand zero term of the balance equation (Eq. 5).

The introduction of only a gradient in leaf area into the balance equation results in, what I call here, a leaf energy–biomass–density relationship or LEBD. Similarly, the introduction of a gradient in available solar radiation into the balance equation

results in an available energy–biomass–density relationship or AEBD. The introduction of both gradients into the balance equation results in the global energy–biomass–density equation or GEBD that applies to equilibrium forest stands worldwide.



## 2.2 Development of the LEBD

The LEBD is developed by introducing light capture in the balance equation, using the leaf area index or *LAI* (leaf area per unit of ground area in $m^2\,m^{-2}$, one-sided for broadleaved trees and the projected leaf area for coniferous trees) and the light extinction coefficient ε (dimensionless) in the adoption of Beer's Law (Monsi and Saeki, 1953):

$$\frac{dM}{M} + \frac{\alpha dN}{N} = -dE_{LAI}/E_{LAI} \tag{7}$$

Here $\alpha$ denotes the scaling exponent that specifically applies to this equation and $-dE_{LAI}$ is the change in the absorbed fraction of the incident radiation $E_{LAI}$ (the incident radiation summed over the growing season, $GJ\,m^{-2}\,yr^{-1}$), which is caused by a small change in *LAI*. $E_{LAI}$ is reduced when it passes through a leaf, and $dE_{LAI}$ is therefore negative. The negative sign is introduced so that the absorbed radiation is positive in the balance equation.

The mathematical relationship between and $-dE_{LAI}/E_{LAI}$ and $dLAI$ is described by the light extinction coefficient, ε, in the adoption of Beer's Law (Monsi and Saeki, 1953):

$$-\frac{dE_{LAI}}{E_{LAI}} = \varepsilon dLAI \tag{8}$$

In this equation, $-dE_{LAI}$ is the fraction ε of the incident radiation $E_{LAI}$ that is absorbed for a small change in the value of *LAI*. Integration over a homogeneous layered canopy results in the well-known equation:

$$E_b = E_o e^{-\varepsilon LAI} \tag{9}$$

where $E_b$ is the below-canopy radiation and $E_o$ is the above-canopy radiation. The consequences of relating ε to the sum of the above-canopy radiation over a growing season instead of radiation intensity values are explained in the Results section of this article. Expressed as a $log_{10}$ value, the equation can be written as:

$$dlog\left(\frac{E_o}{E_b}\right) = 0.4343\varepsilon dLAI \tag{10}$$

and the balance equation becomes:

$$dlogM + \alpha dlogN = 0.4343\varepsilon dLAI \tag{11}$$

Integration results in the next leaf energy–biomass–density relationship or LEBD:

$$log(MN^{\alpha}) = 0.4343\varepsilon LAI + \mu \tag{12}$$

where μ is an integration constant, which is expected to be variable in relation to the global radiation summed over the growing season. Notice that the LEBD is intended to apply to datasets with forests at and below the ceiling area, within the limits of validity set by the application of Beer's Law. The validation of the LEBD against field data of forests results in values of the light extinction coefficient that are tested against literature values (see Results).

## 2.3 Development of the AEBD

The AEBD is developed by the introduction of the available solar energy in the balance equation, using data of the global radiation summed over the growing season $E_{sglob}$ ($GJ\,m^{-2}\,yr^{-1}$). The introduction of $E_{sglob}$ in the self-thinning equation starts with the next extension of the balance equation:





$$d\,logM + \alpha'\,d\,logN = f\,dE_{sglob} \qquad (13)$$

which, after integration, results in the following equation:

$$\log\left(MN^{\alpha'}\right) = fE_{sglob} + g \qquad (14)$$

The scaling exponent $\alpha'$ applies specifically to this available energy–biomass–density relationship or AEBD; $f$ is a regression coefficient, and the intercept $g$ is an integration constant that should be variable in response to changes in the ability of the canopy to capture the available solar energy. This equation applies to a gradient in seasonal global radiation, for instance, due to latitude, under the condition that the capacity of stands to capture energy (dependent on $\varepsilon LAI$) does not change.

**2.4 Development of the GEBD**

When studying global datasets of tree stands, gradients of both $LAI$ and $Esglob$ are combined in a global energy–biomass–density relationship or GEBD:

$$\log(MN^{\gamma}) = 0.4343\varepsilon' LAI + hE_{sglob} + i \qquad (15)$$

where $\gamma$, $\varepsilon'$, $h$ and $i$ are the exponent, the extinction coefficient and regression coefficients, respectively, that apply specifically

to this equation. In fact, the light absorption function of Eq. 15 describes the solar energy absorbed during the growing season $E_{ssol}$ (GJ m$^{-2}$ yr$^{-1}$), which follows from equating $E_{ssol}$ to $E_o - E_b$ and $E_{sglob}$ to $E_o$ in Eq. 9:

$$E_{sglob} = \frac{E_{ssol}}{\left(1 - e^{-\varepsilon' LAI}\right)} \qquad (16)$$

The GEBD is developed to determine the exponent $\gamma$ of $N$ correctly, taking into account the gradients in $E_{ssol}$ (see 'Test of the EBDs'), but the regression coefficients $\varepsilon'$, $h$ and $i$ in the multiple regression equation are attenuated, due to measurement errors

in $LAI$ and $E_{sglob}$ (Aiken and West, 1991). The regression coefficients $\varepsilon$ (Eq. 12) and $f$ (Eq. 14) are expected to be more realistic, of which $\varepsilon$ can be compared with literature values.

**2.5 Test of the model equations**

Five model equations are tested against the data from the compendium of Cannell (Cannell, 1982), including four equations already presented: the mass-density relationship of Eq. 3A, the LEBD of Eq. 12, the AEBD of Eq. 14, the GEBD of Eq. 15

and one new equation I introduce here, the leaf-tree density relationship of Eq. 17:

$$LAI = a\,logN + b \qquad (17)$$

This equation aims to show how leaf area varies with tree density for the interspecific mass-density relationship and is calculated using Ordinary Least Squares regression (OLS) with $N$ as the independent variable, because the estimate of $N$ is much more reliable than the difficult to determine value of $LAI$ (Bréda, 2003).

Together, the five equations aim to give insight into how the scaling exponents and the strength of the linear associations between the variables on the left-hand and the right-hand sides of the EBDs develop by the stepwise introduction of solar energy absorption into the interspecific mass-density relationship of Eq. 3A. The interspecific mass–density relationship of



Eq. 3A is calculated using OLS, with $N$ as the independent variable because $N$ can be counted much more accurately than $M$ in a given area (Sokal and Rohlf, 1981; Niklas et al., 2003). The exponents of the LEBD and the AEBD are determined as the values that maximize the linear association ($r^2$) between the variables on the left-hand and the right-hand sides of the regression equations. The AEBD is calculated using Reduced Major Axis regression (RMA), but the LEBD is calculated using OLS with $LAI$ as the dependent variable, because the measurement error in $LAI$ is expected to be much larger than the error in $\log(MN^\alpha)$. Next, the regression coefficients of the LEBD are determined by recalculating the regression equation to obtain Eq. 12. The exponent of $N$ in the GEBD is calculated as the value that maximizes the linear association ($R^2$) between the variables on the left-hand and the right-hand side of the multiple regression equation. The regression coefficients of the explanatory variables $LAI$ and $E_{sglob}$ are attenuated due to measurement errors in $LAI$ and $E_{sglob}$ (Aiken and West, 1991), but the regression coefficients of $LAI$ and $E_{sglob}$, calculated with the LEBD and AEBD, are more realistic. The statistical significance of all model equations is based on $r^2$ (two-tailed, $Pr<0.05$). Statistical calculations were performed using XLSTAT and the Analysis ToolPak in Excel.

The values of $E_{sglob}$ are mostly determined from the monthly means of daily irradiation and the length of the growing season (monthly mean daily minimum air temperature $T_{min} \geq 0$ ºC, https://www.soda-pro.com/web-services/meteo-data/monthly-means-solar-irradiance-temperature-relative-humidity). For forests in a sea climate (e.g. Belgium, Netherlands, U.K., Japan), where (almost) no $T_{min}$ values are $<0$ ºC, $E_{sglob}$ is determined for the months with the monthly mean air temperature $T_{mean} \geq 5$ ºC. See Supplementary Material for these and additional data used for mountainous areas and the influence of leaf phenology (deciduous species).

**2.6 Selection procedure applied to the forest field and experimental data**

To test the model equations, initially all stands are selected from the forest field and experimental data in Cannell's compendium (Cannell, 1982) with the necessary data on aboveground stand biomass, stand density and leaf area. Next, forests are selected where the biomass and tree density are in a dynamic equilibrium with the absorption of radiant energy, which means that stands subject to notable drought stress and other abiotic disturbances as well as notable animal or human disturbances (e.g. spacing experiments or pruned, severely or recently thinned or coppiced stands) are not included. Also, plantations aged up to 20 years frequently have not reached the dynamic equilibrium established with the EBDs for older stands in the density series. Stands with both broadleaved and coniferous tree species are not selected due to their different light extinction coefficients. Stands with an $LAI$ less than 1.5 were removed to stay within the validity limits set by the application of Beer's Law. In addition, all coniferous tree stands with an $LAI > 10$ were removed because the relationship with $\log(MN^\alpha)$ is unclear. A dataset of 18 stands of *Picea abies* (p. 360-364) was omitted because the 13 stands with the highest $\log(MN^\alpha)$ values did not show a statistically significant relationship with $LAI$.

Stands without accurate LAI data, different values for the dry and wet seasons, or unclear data due to a lack of distinction between trees and other plants such as shrubs and undergrowth were omitted. Stand data only obtained from published





regressions elsewhere were also a reason for omission, especially as reliable estimates of *LAI* are difficult to obtain (Bréda,

2003). The stand selection procedure was also used to achieve normality of the residuals (Shapiro–Wilk and Anderson–Darling

tests) of the regression equations 12, 14 and 15. The selection procedure results in 132 broadleaved tree stands (Supplementary

Table 1) and 67 coniferous tree stands (Supplementary Table 2), which were used to test the five model equations.

## 3 Results

The results of the five model equations applied to the datasets of broadleaved and coniferous tree species separately and

together are presented in Table 1, showing how the scaling exponent and the results for the regression coefficients develop,

when gradients in *LAI* and $E_{sglob}$ are introduced stepwise into the interspecific mass–density relationship of Eq. 3A.

**Table 1. The five model equations applied to globally distributed broadleaved and coniferous tree stands (see Supplementary**
**Material). Pr values are all < 0.0001, except for the Eq. 17 values; 95 % CI values are depicted. The results for the broadleaved and coniferous forests separately and together are depicted in the Fig. 1, 2 and 3 respectively. The adjusted R² values of Eq. 15 in the columns left to right are respectively 0.6559, 0.6428 and 0.6130.**

| Model | Broadleaved stands ($n = 132$) | Coniferous stands ($n = 67$) | All stands ($n = 199$) |
|---|---|---|---|
| Equation 3A: | $r^2 = 0.2702$ F = 48.12 | $r^2 = 0.6182$ F = 105.249 | $r^2 = 0.3048$ F = 86.37 |
| $\log M = \log k -$ | $\beta = 0.22$ CI = 0.16 to 0.29 | $\beta = 0.55$ CI = 0.44 to 0.66 | $\beta = 0.26$ CI = 0.20 to 0.31 |
| $\beta \log N$ | $\log k = 4.09$ CI = 4.03 to 4.15 | $\log k = 3.82$ CI = 3.74 to 3.91 | $\log k = 4.05$ CI = 4.00 to 4.10 |
| Equation 17: | $r^2 = 0.0403$ F = 5.45 Pr = 0.0211 | $r^2 = 0.0035$ F = 0.23 Pr = 0.6334 | $r^2 = 0.0217$ F = 4.37 Pr = 0.0378 |
| $LAI = a\log N + b$ | $a = 0.48$ CI = 0.07 to 0.89 | $a = -0.29$ CI = -1.52 to 0.93 | $a = 0.42$ 95 % CI = 0.02 to 0.81 |
| | $b = 5.57$ CI = 5.17 to 5.96 | $b = 4.76$ CI = 3.82 to 5.70 | $b = 5.42$ 95 % CI = 5.06 to 5.78 |
| Equation 12: | $r^2 = 0.4202$ F = 94.20 $\alpha = 0.34$ | $r^2 = 0.5482$ F = 78.86 $\alpha = 0.52$ | $r^2 = 0.4259$ F = 146.16 $\alpha = 0.35$ |
| $\log(MN^{\alpha}) =$ | $\varepsilon = 0.57$ CI = 0.47 to 0.71 | $\varepsilon = 0.27$ CI = 0.22 to 0.35 | $\varepsilon = 0.50$ CI 0.43 to 0.60 |
| $0.4343\varepsilon LAI + \mu$ | $\mu = 2.72$ CI = 2.38 to 2.94 | $\mu = 3.26$ CI = 3.09 to 3.37 | $\mu = 2.88$ CI 2.66 to 3.04 |
| Equation 14: | $r^2 = 0.5492$ F = 158.35 $\alpha' = 0.56$ | $r^2 = 0.2507$ F = 21.75 $\alpha' = 0.47$ | $r^2 = 0.4473$ F = 159.42 $\alpha' = 0.52$ |
| $\log(MN^{\alpha'}) =$ | $f = 0.30$ CI = 0.26 to 0.33 | $f = 0.14$ CI = 0.11 to 0.17 | $f = 0.25$ CI = 0.22 to 0.27 |
| $fE_{sglob} + g$ | $g = 2.67$ CI = 2.52 to 2.82 | $g = 3.29$ CI = 3.16 to 3.42 | $g = 2.86$ CI = 2.74 to 2.97 |
| Equation 15: | $R^2 = 0.6611$ F = 125.85 $\gamma = 0.47$ | $R^2 = 0.6536$ F = 60.38 $\gamma = 0.50$ | $R^2 = 0.6169$ F = 157.81 $\gamma = 0.43$ |
| $\log(MN^{\gamma}) =$ | $\varepsilon' = 0.16$ CI 0.12 to 0.21 | $\varepsilon' = 0.13$ CI 0.10 to 0.16 | $\varepsilon' = 0.16$ CI 0.13 to 0.19 |
| $0.4343\varepsilon' LAI +$ | $h = 0.15$ CI 0.12 to 0.18 | $h = 0.05$ CI 0.03 to 0.07 | $h = 0.11$ CI 0.09 to 0.13 |
| $hE_{sglob} + i$ | $i = 2.95$ CI 2.82 to 3.08 | $i = 3.37$ CI 3.27 to 3.47 | $i = 3.13$ CI 3.04 to 3.23 |





### 3.1 Model results for the broadleaved dataset

The interspecific mass–density relationship of Eq. 3A applied to the broadleaved dataset (Table 1, Fig. 1) results in a scaling
exponent of β = 0.22, which is lower than the prevailing 0.34 exponent value from the literature (Lonsdale, 1990; Deng et al.,
2012). This can be explained by a reduction in leaf area with decreasing tree density $N$, as shown in the leaf area–tree density
relationship of Eq. 17. The introduction of a light absorption function into the LEBD of Eq. 12, to meet the condition of a
constant rate of energy use (Deng et al., 2012), results in an increase of the exponent from 0.22 to the prevailing 0.34 value
from literature and an increase in $r^2$ from 0.2702 to 0.4202. Note that all the regression coefficients of Eq. 12 in Table 1 are
calculated from the regression results presented in Fig 1(c), Fig 2(c) and Fig 3(c), where LAI is the dependent variable.

It has not been investigated previously that a gradient in available energy summed over the growing season, represented by
$E_{sglob}$, can also influence the slope of the interspecific mass–density relationship. The introduction of a gradient in $E_{sglob}$ in
the interspecific mass–density relationship of Eq. 3A results in the AEBD of Eq. 14, with an exponent of α' = 0.56 and $r^2$ =
0.5492 for the broadleaved dataset, which considerably exceeds the $r^2$ values of the Eq. 3A and 12. The large $r^2$ value as well
as the regression coefficient $f$ = 0.30 of $E_{sglob}$ support the introduction of the AEBD. The exponent of $N$, α' = 0.56, is
considerably higher than the prevailing 0.34 scaling exponent in interspecific scaling and also higher than the 0.50 value of
the self-thinning rule, with consequences for the exponent value in the GEBD.

The value of the scaling exponent in the GEBD (Eq. 15), obtained by introducing both gradients, i.e., $LAI$ and $E_{sglob}$, into the
balance equation (Eq. 5), is expected to lie somewhere between the exponent values of the LEBD and the AEBD. The
introduction of both gradients into the GEBD results in a scaling exponent of 0.47 and a further increase in the strength of the
linear association to $R^2$ = 0.6611 (Table 1, Fig. 1(e)). The regression coefficients of the explanatory variables $LAI$ and $E_{sglob}$
in the multiple regression equation are attenuated, due to measurement errors in the $LAI$ and $E_{sglob}$ (Aiken and West, 1991),
but the regression coefficients of $LAI$ and $E_{sglob}$ in the Eq. 12 and 14 are more realistic, as shown by Eq. 12 where the value
of 0.56 for the extinction coefficient ε can be compared with literature values. The values of ε are usually related to radiation
intensity values ($Jm^{-2}s^{-1}$), instead of the sum of global radiation over a growing season, and vary between 0.40 and 0.66 (White
et al., 2000). These values are independent of the solar elevation angle for broadleaved tree stands (Chen et al., 1997), so the
value of 0.57 is in line with literature values and confirms the applicability of the LEBD to correct for changes in the rate of
energy use in the trajectory of decreasing stand density.





**Figure 1. The five model equations (Table 1) applied to 132 undisturbed broadleaved tree stands distributed globally. Figures 1(a), 1(b) and 1(c): the exponent of $N$ converges from -0.22 to 0.34, when a gradient in $LAI$ is introduced in the mass–density relationship of Fig. 1(a). Figure 1(d): the introduction of a gradient in $E_{sglob}$ in the mass–density relationship results in an exponent of 0.56. Figure 1(e): the introduction of both gradients in the mass–density relationship results in an exponent value of 0.47 in the multiple regression equation and the highest strength of the linear association $R^2 = 0.6611$ (Adjusted $R^2 = 0.6559$).**





### 3.2 Model results for the coniferous dataset

The interspecific mass–density relationship of Eq. 3A applied to the coniferous dataset results in a 0.55 exponent value and an $r^2$ value of 0.6182. The slight exceedance of the 0.50 exponent value is associated with a small (and statistically non-significant) increase in *LAI* with a decreasing stand density, calculated using Eq. 17. The introduction of leaf area into the LEBD of Eq. 12 results in a small reduction of the exponent value to 0.52, which is closer to 0.50, and an $r^2$ value of 0.5482. The reduction of the $r^2$ value compared to Eq. 3A can be attributed to measurement errors in the determination of the *LAI* (Bréda, 2003).

The introduction of a gradient in $E_{sglob}$ in the interspecific mass–density relationship of Eq. 3A results in the AEBD of Eq. 14, with an exponent of α' = 0.47 and $r^2$ = 0.2507 for the coniferous dataset. The coefficient of $E_{sglob}$ in the AEBD of Eq. 14 is $f$ = 0.14 for the coniferous dataset. The low 0.14 value, compared to broadleaved tree species, may be related to a lower competitive ability of many coniferous tree species at higher values of $E_{sglob}$.

The introduction of both gradients, i.e., *LAI* and $E_{sglob}$, in the GEBD (Eq. 15), results in a scaling exponent of 0.50 and an 280 increase in the strength of the linear association between the left- and right-hand side of Eq. 15 to $R^2$ = 0.6536 (Table 1, Fig. 2(e)), close to the $R^2$ value of the broadleaved dataset. The regression coefficients of the explanatory variables *LAI* and $E_{sglob}$ in the multiple regression equation are attenuated, due to measurement errors in the *LAI* and $E_{sglob}$ (Aiken and West, 1991), but calculated separately with the Eq. 12 and 14, of which the value obtained for the extinction coefficient ε in Eq. 12 can be compared with literature values for coniferous tree species. The extinction coefficients of coniferous tree species, calculated 285 to nadir values, are similar to those of broadleaved tree species (between 0.40 and 0.66; White et al., 2000). The low 0.27 ε value for the coniferous forests in Table 1 may be partly due to a greater decrease of the ε value with increasing solar zenith angle (0 directly overhead) due to the generally planophile leaf canopies, compared to broadleaved forests with more random foliage orientation (Chen et al., 1997). The more northerly location of many coniferous forests compared to broadleaved forests, and thus a larger solar zenith angle, also contributes to this effect. The ε value calculated for the coniferous forests is 290 also lower because light absorption in coniferous forests is more affected by shoot clumping, i.e. leaves are more clumped on shoots compared to broadleaved forests, which reduces the light absorption capacity of the canopy (Kim et al., 2011).

### 3.3 Model results for all 199 forest stands

The model test against all 199 forest stands (Fig. 3) doesn't take into account the required distinction between broadleaved and coniferous tree species regarding the value of the extinction coefficient. This results in a value of ε = 0.50 in Eq. 12, which 295 is unrealistic for the coniferous forests in the dataset. Also, the $R^2$ value of the multiple regression equation applied to all 199 stands ($R^2$ = 0.6169), falls below the $R^2$ values of the broadleaved and coniferous datasets separately, making the application of the model equations to all stands less relevant.








**Figure 2. The five model equations (Table 1) applied to 67 undisturbed coniferous tree stands distributed globally. Figures 2(a), 2(b) and 2(c): the exponent of *N* converges from -0.55 to 0.52, with the introduction of a gradient in *LAI* in the mass–density relationship of Fig. 2(a). Figure 2(d): the introduction of a gradient in $E_{sglob}$ in the mass–density relationship results in an exponent of 0.47. Figure (2e): the introduction of both gradients in the mass–density relationship results in an exponent value that converges to 0.5 in the multiple regression equation and the highest strength of the linear association $R^2 = 0.6536$ (Adjusted $R^2 = 0.6428$).**








**Figure 3. The five model equations (Table 1) applied to all 199 undisturbed broadleaved and coniferous tree stands distributed globally. Figures 3(a), 3(b) and 3(c): the exponent of $N$ converges from -0.26 to 0.35, when a gradient in $LAI$ is introduced in the mass–density relationship of Fig. 3(a). Figure 3(d): the introduction of a gradient in $E_{sglob}$ in the mass–density relationship results in an exponent of 0.52. Figure 3(e): the introduction of both gradients in the mass–density relationship results in an exponent value**

**that converges to 0.43 in the multiple regression equation and the highest strength of the linear association $R^2$ = 0.6169 (Adjusted $R^2$ = 0.6130).**



## 4 Discussion

In this article the long-standing debate on the value of the scaling exponent in interspecific mass–density scaling of plants is addressed by replacing the scaling coefficient with a light absorption function, distinguishing between two premises. The first

common premise in self-thinning theories is that the scaling coefficient represents a limiting use of resources supplied to an area at a fixed rate, here assumed to be solar radiation (Deng et al., 2012), which is examined by introducing $LAI$ into the light absorption function. The second alternative premise examines the possibility that the scaling coefficient depends not only on the rate of solar energy use, but also on the sum of solar radiation over the growing season, which is investigated by including both $LAI$ and $E_{sglob}$ in the light absorption function.

The investigation of the first premise begins with a comparison of the results of the three equations 3A, 17 and 12 applied to the broadleaved and coniferous dataset separately and together. The exponent values β = 0.22 and β = 0.55 obtained with the mass–density equation (Eq. 3A) for the broadleaved and coniferous dataset respectively differ considerably from one another and from the prevailing 0.34 exponent value of $N$ obtained for large datasets of plants (Deng et al., 2012). This deviation is not uncommon as shows in the exponent values of the much larger dataset of 1350 natural forests pictured in Fig. 4. The

predominantly coniferous boreal/alpine forests show a much higher exponent value β than the predominantly broadleaved (sub)tropical and temperate forests (see also Li et al., 2005, 2006), which corresponds with the β values in Table 1.

The deviation of the broadleaved dataset from the common 0.34 exponent value (Deng et al., 2012) can be explained by the density-dependent gradient in $LAI$ (Eq. 17), resulting in a 0.34 exponent value of $N$ obtained with the LEBD (Eq. 12). However, a density dependent gradient in $LAI$ is hardly recognizable in the coniferous dataset, which results in a small decrease of the

exponent from 0.55 to 0.52 in the LEBD. The results obtained with the LEBD are statistically significant and credible because the values of the extinction coefficient ε for the broadleaved and coniferous forests separately correspond with literature values. The value of the exponent for all stands together is 0.35, but the distinction between the broadleaved and coniferous dataset has to be preferred.

The investigation of the second alternative premise is based on the observation that stand density increases with decreasing

latitude, while the total aboveground biomass remains constant (Enquist and Niklas, 2001) or increases (Pan et al., 2013), so $\log(MN^{\alpha'})$ in Eq. 14 should increase with decreasing latitude and an increase in $E_{sglob}$. This expectation is confirmed by Eq. 14 applied to the broadleaved and coniferous forests separately and together (Table 1) and is also visible in the intercepts of the mass–density relationships in Fig. 4, where the log$k$ values increase in the order of boreal/alpine, temperate and (sub)tropical forests.

Note that the log$k$ values of the boreal/alpine, the temperate and (sub)tropical and all forests combined in Fig.4 are lower than those for the coniferous, the broadleaved and all forests combined in Table 1, respectively, due to the large proportion of forests with a potential evapotranspiration considerably higher than the annual rainfall in dataset S1 of Deng et al. (2012), indicating substantial drought stress. Drought stress limits the ability of forests to maximize the dynamic equilibrium of biomass and tree density, given the available solar energy summed over the growing season.





**Figure 4. Relationships between total aboveground living tree biomass, *M*, and tree density, $N_{crit}$, on logarithmic axes (Eq. 3A), for 1350 natural forests (Deng et al. 2012, dataset S1, data Luo and Cannell): 477 (sub)tropical forests, 608 temperate forests and 265 boreal/alpine forests. A single OLS regression line fitted to all the 1350 forests has a slope of -β = -0.21 and an intercept of 3.80, r² = 0.09. The use of $\bar{M}$ values (Eq. 3) results in r² values of 0.78, 0.85, 0.78 and 0.86 for respectively all, the (sub)tropical, the temperate and the boreal/alpine forests, the slopes -β' are equal to -β - 1, but the intercepts remain the same.**

The inclusion of both *LAI* and $E_{sglob}$ in the light absorption function of the GEBD, applied separately to the broadleaved and coniferous tree species, generates the most important results of this investigation. The GEBD shows the strongest linear associations of all five model equations and an exponent of *N* that converges to 0.47 for the broadleaved and 0.50 for the coniferous forests. This means that the *LAI* and $E_{sglob}$ together, i.e., the absorption of solar energy $E_{ssol}$, correct the exponent in the interspecific mass-density relationship for gradients in the energy use over the growing season, resulting in an exponent that is almost the same for broadleaved en coniferous forests, respectively close and equal to 0.50.

This throws new light on the difference between the 0.34 exponent in interspecific mass–density scaling of large compilations of plant communities and the 0.50 exponent in intraspecific scaling as the upper boundary of datapoints of monospecific even-aged plant stands. The results for forests suggest that the exponent in the interspecific mass–density relationship depends not



only on the premise of a fixed rate of solar energy absorption, but also on the sum of solar energy absorbed over the growing season. The light absorption function in the GEBD corrects for the gradient in $E_{ssol}$ in the interspecific data, resulting in an exponent of $N$ that converges to 0.50, equal to the self-thinning rule in intraspecific scaling, where the datapoints are often obtained from stands in the same region, resulting in negligible gradients in $E_{sglob}$ at the upper boundary of datapoints. The energetic explanation for the difference in the scaling exponent between interspecific and intraspecific scaling may be of

interest for the interpretation of scaling studies on other plants and even animals, if gradients in the sum of energy use, in addition to the rate of energy use (e.g., West et al., 1997; Enquist et al., 1998), also determine the slope and intercept of the mass-density relationship. The GEBD calculates the scaling exponent correctly, but the regression coefficients in the GEBD are attenuated, due to measurement errors in the $LAI$ and $E_{sglob}$. This will be addressed in a forthcoming paper by analyzing the structure of the underlying data, using time series and intraspecific density series of forests with little or no biotic, abiotic

and human disturbances.

The GEBD developed in this article describes the mathematical relationship between observations, without a further description of the underlying processes, which is very different from the previously cited mechanistic model approaches, based on mechanisms such as xylem transport, biophysical packing or ecological field theory. These models treat self-thinning as a competition process driven by rates of resource supply, whereas here the focus is on the gradient in available energy summed

over the growing season, which also determines the position of self-thinning lines. The model approach developed in this article aims to bridge the gap between empirical self-thinning models, which achieve high precision within their domain of development and parameterization, and process-based competition models, which offer a generalized framework but suffer from a lack of precision in long-term predictions for scenario analysis due to compounding of errors (Franklin et al., 2009). Here the dynamic equilibrium between forest biomass and tree density on the one hand and the absorption of solar energy

$E_{ssol}$ on the other has been assessed for interspecific density series of relatively undisturbed forests. A method that will also be applied to intraspecific density series of forests in a forthcoming article, thus contributing to better predictions of forest structure, which is of importance for the inclusion of vegetation demographics in Earth System Models (ESMs; Fisher and Serbin, 2017).

For now, it's unclear to what extent the GEBD can be used to predict the impact on forest structure of an increase in $E_{sglob}$

due to climate-induced longer growing seasons, as the gradients in $E_{sglob}$ go together with shifts in the species composition between the forest communities of interest. Self-thinning lines of long-term trial plots of Norway spruce, European beech (Pretzsch et al., 2014) and Pinus sylvestris (Toraño Caicoya et al., 2024) do not show trends in time due to climate change-induced longer growing seasons, only the growth rate increases. However, the intercepts of the self-thinning lines of long-term trial plots of Pinus sylvestris in Europe increase with decreasing latitude (Toraño Caicoya et al., 2024) and with increasing

$E_{sglob}$, which is associated not only with a longer growing season but also with an increase in the intensity of solar radiation. This may lead to an increase in the intercept that is consistent with the results of light experiments (see Introduction).



In natural stands, where individuals of all age classes are present simultaneously, a steady state may ultimately be reached where the net production is equal to the losses, and changes in $M$ tend to be zero. This means that the energy budget, as the sum of the ingoing and outgoing energy fluxes, is also zero, and it is not appropriate to specify the energy term $\log k$ in the mass-density relationship. However, from a thermodynamic point of view, the entropy production is as important as energy. Following a further determination of the regression coefficients in the GEBD, the entropy production can be introduced into the GEBD, using the strong linear association with $E_{ssol}$ (e.g., Aoki, 1987, 1989; Brunsell et al., 2011), which allows the development of a physically correct thermodynamic equation.

The empirical pattern of forest mass–density relationships fits into a thermodynamic framework, because thermodynamics can deal with the mathematical relationship between observations, without (sub)models of the underlying processes. Although the model approach in this article and mechanistic models represent different scales of inquiry and different scientific approaches, they can inform one another to obtain more insight into the common processes underlying these theories (Price et al., 2010).

**Competing interests**

The author declares no conflicts of interests.

**Acknowledgements**

The author thanks Prof. Anne Verhoef of the University of Reading, Reading, United Kingdom for comments.

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
