# Peer review of "Variation of the interspecific forest mass-density relationship along gradients of leaf area and global radiation"

_EGUsphere, 2024_

## Author Comment (AC1)

Author response to reviewer 1, Pablo C. Salazar

Reviewer comments in black
Author responses in blue

The creation of an improved power law equation that incorporates available radiation and leaf metrics is necessary, given the advancement in technology to estimate leaf area index (either through field measurements or remote sensing techniques). This paper provides new insights into how we build mass-density relationships in forest ecology and includes new variables to improve our biomass models.

I thank the reviewer for her helpful comments and appreciation of the contribution of this work. In the responses below I set out the approach to address the comments.

However, the document requires substantial revision before it can be accepted. In particular, more details about data collection methods and the main objectives of the manuscript need to be clearly stated.

The Yoda power law discussion creates confusion, as it seems this can only be applicable in monospecific forest plantations with well-established designs. At some points in the introduction, the manuscript addresses this as a real limitation of the equation (e.g., Lines 47 or 57). However, in other sections, the text seems to dismiss these limitations simply because the logic behind the equation appears intuitive (Line 70). This may create the misconception that these equations could predict forest biomass globally. The manuscript later acknowledges its limitations at the end of the introduction when indicating that only "undisturbed stands" will be included in the model testing. Therefore, it would be wise to clearly distinguish between the actual application of the equation and the broader theoretical implications.

The objective of the manuscript is not clearly stated in either the introduction or the abstract. As I understand it, the objective is to develop a model for the mass-density relationship that uses a light absorption function instead of a normalization constant. This should be explicitly stated in the document for better clarity.

The abstract and introduction are substantially revised to clearly state the objectives and clearly distinguish between the actual application of the equation and the broader theoretical implications. The text has been rearranged. The item about the Yoda power law has been replaced to the Discussion. Equation 1 has been removed, which means that all equations in the text have been renumbered. Two new references are included. The full text of the abstract and introduction are presented below. The most significant changes to the text are indicated in bold letter type.

Abstract. **Forest stand mass scales with varying values of the exponent of tree density for compilations of forest communities on different spatial scales**, being the slope of the regression line in a log–log plot, where the intercept is a normalization constant reflecting the assumption of a constant rate of energy use by the species and environments involved. **The objective of this article is to develop a model, using a light absorption function instead of a normalization constant,** to investigate how the interspecific forest mass-density relationship varies along spatial, largely latitudinal gradients of leaf area and the sum of global radiation over the growing season for relatively undisturbed forests **distributed across biomes in the Northern Hemisphere**. The test of the model shows the highest explained variance when both gradients are included in the light absorption function, meaning that the exponent is determined not only by the rate but also the sum of energy use over the growing season. The exponent of tree density converges to 1/2, which deviates from an expected 1/3 value based on mass-density scaling of large compilations of plant communities on a continental or global scale. The 1/2 value corresponds

with the so-called self-thinning rule that applies to the self-thinning line constructed as the upper boundary of mass–density points for monospecific even-aged plant stands, where gradients in energy use can be neglected. The results demonstrate the appropriateness of replacing the normalization constant with a light absorption function, suggesting a thermodynamic interpretation that may be of interest to other plants and even animals when gradients in energy use similarly affect the intercept and slope of the interspecific mass–density relationship.

1 Introduction

**In order to predict climate-induced changes in forest ecosystem processes at different spatio-temporal scales, it is crucial to understand how available resources (i.e. energy, nutrients and water) interact with forest structure dynamics (i.e. size and density) and forest functions that affect carbon stocks and fluxes. The allometric relationship between the average live aboveground biomass per tree in an area $\overline{M}$ (g) and the tree number in that area $N$ (m$^{-2}$) links forest structure to forest functions for relatively dense stands, with the underlying rationale that an increase in the average biomass per tree is associated with tree mortality due to competition for the available resources (Westoby, 1984; Yu et al., 2024). However, the mathematical expression does not include resource variables, which makes it difficult to understand the interaction of the mass-density relationship with (gradients in) resource use in density series of stands at different spatial scales. Here, the allometric mass-density relationship is adjusted by including the resource variable light and testing the model for interspecific density series of relatively undisturbed forests with negligible drought stress at broad spatial scale, starting from the equation**:

$$\overline{M} = kN^{-\beta'} \tag{1}$$

where -β' is scaling exponent and $k$ the scaling coefficient or normalization constant that adjusts the general relationship across environments and species. For statistical reasons $\overline{M}$ is written as a function of $N$, because the number of trees in an area can be determined much more accurately as the independent variable in a linear regression model where the general mass-density equation is plotted on logarithmic axes:

$$\log\overline{M} = \log k - \beta'\log N \tag{2}$$

where -β' is the slope of the regression line and log$k$ the y-intercept.

The exponent -β' was for a time thought to converge to -3/2 (the 'self-thinning rule', Yoda et al.,1963; Westoby, 1984; White, 1985), but closer examination revealed that the slope and intercept vary considerably when the equation is applied to intraspecific and interspecific density series of plant stands at different spatial scales (Weller 1987a, 1987c; Lonsdale, 1990). Only for large interspecific datasets of plant communities at continental and global scales, where $N$ is the critical density of maximally packed individuals where all resources are used, an exponent value of -4/3 (or -3/4 as the exponent of $\overline{M}$, using $\overline{M}$ as the independent variable in Eq. 1) is reasonably well founded (Deng et al., 2012; White et al., 2007). **However, the generalisation of this value to mass-density relationships of tree-dominated communities of boreal, temperate, subtropical and tropical biomes separately seems to be unwarranted, as shown for the Chinese Forest Biomass Dataset (Luo, 1996, Li et al., 2006).**

Mechanistic models based on geometric, allometric and dynamic growth arguments (e.g., Weller, 1987b; Adler, 1996; Enquist et al., 1998; Li et al., 2000; Deng et al., 2012) reproduce the general mass-density equation well, with plausible values of -β', but it remains problematic to predict the slopes of empirically obtained mass-density

relationships. In most of these models, the constant $k$ is assumed to be a measure of a constant whole stand rate of resource use, including light (e.g., Westoby, 1984; White et al., 2007; Deng et al., 2012), with variations between mass-density relationships that can be stronger than variations in the exponent (Deng et al., 2006; Dai et al., 2009). However, the variation in $k$ has received relatively little attention (Dillon et al., 2019), although the estimate of the variation is correlated with the estimate of $-\beta'$ in Eq. 2 (Westoby, 1984).

**The objective of this investigation is to develop a model that uses a light absorption function instead of the normalization constant, with the purpose of incorporating gradients in total energy use in the general mass-density equation. The model is tested for interspecific density series of relatively undisturbed forests, where energy use is expected to be in a dynamic equilibrium with the living aboveground biomass and tree density. Two common assumptions are investigated with the new model: 1) the assumption that the rate of energy use is constant, while stand leaf area varies with stand age (Ryan et al., 1997; Holdaway et al., 2008) and 2) the assumption that the rate of energy use drives self-thinning (e.g. Deng et al., 2012), while gradients in the sum of energy use over the growing season can also be considered. The impact of the first assumption emerges in** light experiments on monospecific even-aged plant populations, comparing self-thinning trajectories for different but constant levels of illumination. The intercept of thinning lines is lowered with increasing shade, while the scaling exponent at each level of shade is maintained at a value of approximately -3/2, except for populations grown under deep shade (Hiroi and Monsi, 1966; Lonsdale and Watkinson, 1982, 1983; Hutchings and Budd, 1981; Westoby and Howell, 1981; Westoby, 1984). This suggests that reduced light absorption due to a lower total leaf area, rather than increased shade, also results in a lower intercept of the thinning line, but only a different slope if light absorption is not constant throughout the trajectory of self-thinning. The effect of gradients in total leaf area, and thus light absorption, on the exponent of $N$ in the forest mass–density relationship is investigated by developing a light absorption function that includes total leaf area, **measured by the Leaf Area Index or $LAI$ ($m^2 m^{-2}$) of the forest stand**, to replace the normalization constant k.

The investigation of the second assumption is intuitively justified by the finding that tree density increases with decreasing latitude to approximately 25°, while the total aboveground biomass is supposed to be constant (Enquist and Niklas, 2001) or increases, apart from the spatially restricted temperate rainforests (Pan et al., 2013). Therefore, the light dependent constant $k$, equal to $\bar{M}N^{\beta'}$ according to Eq. 1, increases with decreasing latitude, together with an increase in the sum of available solar energy use over the growing season, here investigated by including this gradient in a light absorption function, separately from and in addition to leaf area. The leaf area and the available solar radiation over the growing season are introduced stepwise in the light absorption function that replaces the scaling coefficient $k$, to correct for gradients in the total energy use of forests. This results in three energy–biomass–density relationships or EBDs that are used to examine how the interspecific mass–density relationship of relatively undisturbed forests varies along spatial, largely latitudinal gradients of leaf area and available solar energy separately and together. In addition to the three EBDs, the bivariate mass–density relationship and the relationship between leaf area and stand density are calculated, giving insight into how the slope of the mass–density line relates to the gradient in total leaf area. The analysis focuses on the development of the exponent of $N$ because this is a central issue in the debate on the mathematical form of the mass-density equation applied to forests. In addition, we examine the extent to which the regression coefficients support the conclusions with respect to the development of the exponent in the EBDs. **In a follow-up paper the model will be applied to density series of neighbouring stands with similar species composition, which contributes to a**

**better understanding of the relationship between interspecific and intraspecific scaling for relatively undisturbed forests and of the interaction of resource use with forest structure and forest functions for separate species and habitats.**

In this introduction is referred to the allometric mass–density relationship using $\bar{M}$, conform most cited studies, but the model development in this paper will be based on equations written in terms of the total living aboveground biomass $M$ (g m$^{-2}$), because the calculation of $\bar{M}$ as $M/N$ from the available field data introduces artificially inflated correlations (Weller, 1987a). The use of $M$ leads to an exponent $-\beta = -\beta' + 1$, while $k$ stays the same (see Methods). The EBDs assume a constant regime of light absorption over the years that is long enough to establish a dynamic equilibrium with the aboveground living biomass and tree density. Human, biotic and abiotic disturbances like thinning (not self-thinning), insect diseases and drought stress can lead to deviations in leaf area from this dynamic equilibrium, due to functional responses of forests to disturbances (Jump et al., 2017). Therefore, only relatively undisturbed stands are included in the test of the model equations. The model equations are applied to the field data of forest biomass and density in the compendium of Cannell (1982) that comprises standardized tabulations of field and experimental data of forests of approximately 600 reports worldwide.

**New references:**

Hamilton, N.S., Matthew, C. and Lemaire, G. (1995). In defense of the -3/2 boundary rule: A re-evaluation of self-thinning concepts and status. Ann. Bot. 76, 6, 569–577, https://doi.org/10.1006/anbo.1995.1134. 1995.

Yu, K., Chen, H.Y.H., Gessler, A., Pugh, T.A.M., Searle, E.B., Allen, R.B., Pretzsch, H., Ciais, P., Phillips, O.L., Brienen, R.J.W., Chu, C., Xie, S., Ballantyne, E.P., 2024. Forest demography and biomass accumulation rates are associated with transient mean tree size vs. density scaling relations. PNAS Nexus 3: 1–9, https://doi.org/10.1093/pnasnexus/pgae008, 2024.

The final five paragraphs of the discussion focus on limitations and scope of the results. However, the potential applications of the results are missing from the last paragraph. It would be better to conclude the discussion with the manuscript's conclusions, though this may be challenging without a clear explanation of the objectives.

The first paragraph and the first three of the last five paragraphs of the discussion have been revised. The new text of the first paragraph is presented below. The first three paragraphs of the last five paragraphs have been rewritten and are presented also. The last two paragraphs have been maintained. The most significant changes to the text are indicated in bold letter type.

The first paragraph of the discussion:
**In this article the interaction of resource use (i.e. light) and stand structure dynamics of relatively undisturbed forests at broad spatial scales is investigated by replacing the normalization constant in the forest mass-density scaling relationship with a light absorption function, distinguishing between two assumptions. The first common assumption in interspecific mass-density scaling relationships** is that the scaling coefficient represents a limiting use of resources supplied to an area at a fixed rate, here assumed to be solar radiation (Deng et al., 2012), which is examined by introducing $LAI$ into the light absorption function. **The second assumption** examines the possibility that the scaling coefficient depends not only on the rate of solar energy use, but also on the sum of solar radiation over the growing season, which is investigated by including both $LAI$ and $E_{sglob}$ in the light absorption function. **In the investigation of the assumptions, using the data of 199 relatively undisturbed forest communities selected from Canell (1982), a comparison is made with a much**

larger dataset of 1350 natural forests obtained from Deng et al. (2012), where $LAI$ data are generally missing (see section 2.7).

The first three of the last five paragraphs of the discussion have been replaced by the text (most significant changes in bold letter type):

The GEBD calculates the scaling exponent correctly, but the regression coefficients in the GEBD are attenuated, due to measurement errors in the $LAI$ and $E_{sglob}$ (Aiken and West, 1991). This will be addressed in a forthcoming paper by analysing the structure of the underlying data, using intraspecific time and density series of forests with little or no biotic, abiotic and human disturbances. The use of EBDs to analyse intraspecific data will also provide more information on the theoretical aspects and potential applications of the EBD model discussed below.

**The results of the GEBD suggest that deviations from the 0.50 exponent in bivariate interspecific scaling of relatively undisturbed forests at broad spatial scales can be attributed to gradients in the sum of energy use over the growing season. This is consistent with the classic 'self-thinning rule', which was once thought to predict a 0.50 exponent of tree density for interspecific and intraspecific density series (Westoby, 1984). Currently the rule is considered to describe only the upper boundary of data points in a $\log M - \log N$ plot of crowded even-aged monospecific plant populations, where the datapoints are often obtained from stands in the same region (Hamilton et al., 1995). Here, the gradients in *LAI* and $E_{sglob}$ are negligible small, so the GEBD is expected to converge to the mass-density relationship of Eq. 2A, with the light absorption function equal to $\log k$, representing the energy use at the ceiling leaf area. The absence of gradients in energy use is much less likely in other types of intraspecific density series, which argues in favor of applying the EBDs. The application of EBDs to intraspecific mass-density relationships can provide more insight into the relationship between interspecific and intraspecific scaling.**

**The EBDs show how the interspecific forest mass-density relationship interacts with the resource energy (i.e. light), but the question is how it interacts with the resource nutrients for relatively undisturbed forests with negligible drought stress. The *LAI* is central to answering this question, because it can be seen as a proxy of functional responses of forests to the resource availability of e.g. energy and nutrients (Jump et al., 2017). Hamilton et al. (1995) reasons that competition for light, rather than nutrients, causes mortality and an increase of leaf area shifts the limiting thinning line upwards. An increase of the nutrient supply increases the rate of progression of the thinning line, without changing the position of the line (Yoda et al., 1963; White and Harper, 1970). The evidence that nutrients also alter the position of the line (Westoby, 1984; Morris and Myerscough, 1991) is ambiguous. If nutrients change the position of the line, there may be an indirect effect on the leaf area and thus the capacity of plant stands to absorb light (Hamilton et al., 1995). This reasoning suggests that the *LAI* in the light absorption function links both energy and nutrients to the forest mass-density relationship.**

**The exponent -β in the general mass-density relationship (Eq. 2A) reflects the ratio of the relative growth rate and the relative mortality rate of the forest stand (RGR/RMR ratio, Eq. 3). A more negative value, or steeper slope in a $\log M - \log N$ plot, implies a larger RGR/RMR ratio, i.e. the biomass accumulation rate increases when the mortality rate is constant, or the mortality rate decreases when the biomass accumulation is constant. As the exponent becomes less negative, the reverse reasoning applies (Yu et al., 2024). The bivariate mass-density relationship is not suitable for determining whether the variation in exponent values is associated with light and nutrient use or other environmental drivers, but the GEBD is, because gradients in light and nutrient use are included in the light absorption function, not in the exponent value that converges to 0.50. Deviations from the 0.50 exponent may indicate other environmental or vegetation property changes like, for instance, disturbances that prevent the forest structure to be in a dynamic**

equilibrium with the use of energy and nutrients as formulated with the light absorption function. In this way, the EBDs can provide further information on the interaction of changing environmental conditions with forest demography and carbon sequestration at different spatio-temporal scales, using the advancement in technology of field measurements and remote sensing techniques to estimate leaf area index and other stand indices like tree density and size. A better insight in *LAI* and $E_{sglob}$ as predictors of stand structure and growth (see also Parker, 2020), together with high-resolution satellite or aerial remote sensing data at broad spatial scales, contributes to the inclusion of size-structured forest demographic models in Earth System Models (ESMs; Fisher and Serbin, 2017).

The EBDs are also of importance for forestry practice, where self-thinning lines are used with the idea that all stands should eventually approach and track along the same maximum size-density relationship (MSDR) boundary for a particular species/region combination (Vanderschaaf and Burkhart, 2007). The EBDs allow investigating of how spatio-temporal variation in MSDRs across regions and species relates to gradients in light (and nutrient) use, helping to reduce the empirical nature of MSDRs and hence errors in growth and yield models constrained by MSDR boundary lines (Vanderschaaf and Burkhart, 2007).

In the interspecific GEBD, gradients in $E_{sglob}$ are associated with shifts in species composition between the forest communities involved, making it difficult to predict the impact of climate-induced longer growing seasons on forest structure. Application of the EBD model to long-term trial plots is recommended, provided reliable *LAI* data are available. Self-thinning lines of Norway spruce, European beech (Pretzsch et al., 2014) and Pinus sylvestris (Toraño Caicoya et al., 2024) do not show trends in time due to climate change-induced longer growing seasons, only the growth rate increases. However, the intercepts of the self-thinning lines of Pinus sylvestris in Europe increase with decreasing latitude (Toraño Caicoya et al., 2024) and with increasing $E_{sglob}$, which is associated not only with a longer growing season but also with an increase in the intensity of solar radiation. This may lead to an increase in the intercept that is consistent with the results of light experiments (see Introduction).

Line 123: Where did you obtain the leaf area index data? It would be surprising if it came from the Cannell database. This should be detailed in the materials and methods section as it is not a standard variable. In line 203, you mention obtaining it from the Cannell database, but could you elaborate on how this variable was measured? If obtained through remote sensing, this should be explicitly stated, as mixing estimated and field data could be a limitation of your work. After reviewing the supplementary material, the LAI values seem unusually high. Typically, these values range from 4 to 6, meaning the canopy cover is 4-6 times higher than the projected ground area. With your reported LAI values, it's surprising that any light reaches the soil. As someone with field experience measuring LAI, values above 3 indicate dense forest canopies. Your database shows no values under 1, suggesting your equation may only be applicable to very dense forests or forest canopies. Please explain how the LAI measurements were conducted.

Indeed, as indicated in line 209: Stands with an LAI less than 1.5 were removed from the database to stay within the validity limits set by the application of Beer's Law. Also, most of the stands in the database have relatively high LAI values. I think the high LAI values of most stands are due to the selection of relatively undisturbed forests with negligible drought stress. The application of the equations to intraspecific density series of neighbouring tree stands with a corresponding species composition in a follow-up paper can give more information on the validity range of the light absorption function (see Discussion). The following text with respect to the measurement methods is inserted between the two sentences in line 209:

**The data on $LAI$ of the selected stands were measured using litter traps, allometry based on sample trees or destructive methods. An example of the use of litter traps is the deciduous woodland in Sweden (p. 222 Cannell, Supplementary Table 1), where the $LAI$ value was averaged over a three-year measurement period. An example of the use of allometry are Pinus resinosa plantation forests in the U.S.A., New York (p. 314 Cannell, Supplementary Table 2), with 5 sample trees per plot in 4 plots and 3 sample trees in one plot. The $LAI$ values were derived from regressions on breast-height diameter D. Examples of destructive methods are the Abies alba forest and deciduous forest in Czechoslovakia (p. 57 Cannell, Supplementary Table 1, 2). The $LAI$ values were derived by multiplying the means of 5 sampled dominant trees, 5 co-dominants and 5 subdominants by the numbers of trees in each of these classes.**

Line 49: Should this have an additional reference, or is this information contained in (Weller, 1987a)? This sentence is not included in the new text.

Line 61: The manuscript mentions "leaf area," but it seems to refer to "total leaf area," as individual tree leaf area doesn't vary significantly. This distinction should be clarified in the introduction. While the authors later refer to leaf area index as a measure of total leaf area for their calculations, greater precision in terminology would be helpful. This is clarified in the previously presented new introduction.

Line 131: Is the first "and" necessary? "and" will be removed

Line 159: Consider mentioning hillshade as a factor influencing seasonal global radiation. After latitude has been inserted 'and hillshade'

Line 217: Since most of your data comes from broadleaved trees, this should be mentioned as a limitation. At the end of line 218 is added: Most of the data are from broadleaved forests, which can be considered a limitation.

Table 1: Consider adding a column at the beginning with equation names (e.g., "AEBD model" or "LEBD model"). In the first column the name will be added. For instance: 'Equation 14' becomes 'AEBD Eq. 13'. Note that Eq. 1 is removed, so Eq. 14 becomes Eq. 13.

Line 231 & 271: How can you explain the lower scaling exponent due to reduced leaf area if this variable isn't used in the equation? This appears to be an educated guess rather than a direct result from your model. Consider adding "Possibly" at the beginning of the sentence in Line 271. Line 231: 'This can be explained by a reduction in leaf area with decreasing tree density N' will be replaced by 'A plausible explanation is a reduction in leaf area with decreasing tree density'. Line 271: 'Possibly' will be added at the beginning of the sentence.

Line 234: Is it valid to compare scaling exponents when the equations are fundamentally different? The mass-density Eq. 2A can be considered as a special case of EBDs, namely when gradients in $LAI$ and $E_{sglob}$ are negligible. So, these equations can be compared. See also the new text of the Discussion (second paragraph of the new text that replaces three of the last five paragraphs).

Line 235: Figures should appear in the document in the order they are referenced. Asking readers to check figures 1c, 2c, and 3c makes the flow difficult to follow.
The sentence 'Note that the regression coefficients of Eq. 11 are calculated from the regression results presented in Fig 1(c), Fig 2(c) and Fig 3(c), where LAI is the dependent variable' will be included in the legend of Table 1.

Line 287: A map showing plot locations (with different colors for each forest type) is necessary, especially given that you explain results based on geographic distribution.
This map will be included in the Supplementary Material.

Line 349: A "second alternative premise" is mentioned, but this doesn't appear to be an original objective of the paper. Again, the objectives should be clearly stated in the introduction. Furthermore, it's surprising that a proper latitudinal analysis isn't included in the methods or results, yet a new figure appears in the discussion.
In the new 'Introduction' and 'Discussion' presented before two 'assumptions' instead of 'premises' are considered. The latitudes of the 199 selected forests with LAI data are considered at the end of section 2.6. The next sentences are added:
The latitude of the broadleaved forests varies between ca. 2° and ca. 56° N, with an average latitude of ca. 36° N. The latitude of the coniferous forests varies between ca. 32° and ca. 58° N, with an average latitude of ca. 41° N. The coniferous dataset includes thirteen forests with a latitude between 1500 and 2740 m NAP.

The latitudes of the boreal/alpine, the temperate and the (sub)tropical dataset are considered in the new section 2.7, presented below:
**2.7 Notes on data from Deng et al. (2012)**
Dataset S1 from Deng et al. (2012) is used to obtain an indication of whether the selection of 199 forest stands with $LAI$ data from Cannell (1982) is representative of a larger selection of forests for which $LAI$ data are generally missing (see Discussion). The dataset includes a subset of 1350 natural forests from the Chinese Forest Biomass Dataset (Luo, 1996) and the Cannell (1982) compendium, of which 1109 forests are from the Luo database and 241 are from Cannell (1982). The Luo database includes 6 forest biomes across the entire country. The 6 biomes are: boreal/alpine, temperate deciduous broadleaved, temperate coniferous, subtropical evergreen, subtropical coniferous and tropical rainforest/monsoon forest. Here, the 1350 forests are grouped into a boreal/alpine, temperate and subtropical/tropical group of forests, which are expected to differ in $E_{sglob}$ due to latitudinal and altitudinal gradients. The interspecific forest mass-density relationship (Eq. 2A) is applied to compare these groups with the 199 forests including $LAI$ data (see Discussion). The equation is also applied to all 1350 forests together. The boreal/alpine subgroup includes 252 coniferous and 13 broadleaved forests, the temperate subgroup 389 broadleaved and 219 coniferous forests and the (sub)tropical subgroup 335 broadleaved and 142 coniferous forests. The latitudes of the boreal/alpine subgroup (between ca. 27°-64° N) and the temperate subgroup (between ca. 27°-56° N) are about the same on average (ca. 40° N), but $E_{sglob}$ of the boreal/alpine subgroup is smaller on average due to the relatively short growing season of 166 alpine forests with a latitude between 27°-40° N and an altitude of ca. 2000 m NAP or higher. So, the gradient in energy use is also related to an altitudinal gradient. The average latitude of (sub)tropical forests (between ca. 7°-38°) is about 28°. This includes forests in the southern hemisphere also.

Line 357: The location of dataset S1 from Deng et al. (2012) and its relationship to your equation in Table 1 needs clarification. The data, analysis, and reasoning behind Figure 4 are completely missing

from the document. It's problematic that the largest figure in the manuscript appears without proper context or explanation.

The data, analysis and reasoning are now included in the Introduction, section 2.7 and the Discussion.

---

## Author Comment (AC3)

Author response to reviewer 2

Reviewer comments in black
Author responses in blue

I first want to apologize for the long delay in reviewing this manuscript. When I started reading it, I already realized why it is difficult to review this MS - it in principle deals with a relevant topic, it could have interesting insights, but the way it is organized and written, it is very difficult for me to follow and make sense out of it, sometimes things are being repeated, and overall, it misses a focus on what the goal of the whole manuscript is.
What is actually the hypothesis that is being evaluated? Why is this hypothesis relevant? These aspects do not seem to be addressed at all, rather, different fitting equations are presented, and the results of the fitting exercise are described. The data sources are not well described, the climatological forcing comes from an obscure data source that I have never heard before. And in my view, there is too much description of statistical fitting and too little scientific explanations/interpretation.

I thank the reviewer for the helpful comments and appreciation that the manuscript could have interesting insights.

I propose to rewrite and reorganize the manuscript as explained below. First, I interpret the relevant sections of the manuscript, indicated by line numbers, regarding the reviewers' comments. Second, I present the new text in the letter type of the original manuscript.

Regarding the reviewer's comments 'in my view, there is too much description of statistical fitting and too little scientific explanations/interpretation': I hope that the proposed changes to the text, particularly in the introduction and discussion sections, will provide the desired scientific explanations/interpretations and less focus on statistical descriptions. Also, the relatively large focus on statistical aspects compared to scientific explanations is related to the thermodynamic nature of the EBD model. As is possible with thermodynamic models, the EBD model describes the mathematical relationship between observations without explanatory (sub)models of the underlying processes (see the chapter Discussion).

Regarding the reviewer's comment, 'The data sources are not well described': this response provides additional information regarding my response to Reviewer 1.

At the end of this response the new references compared to the preprint of 9 October 2025 are added.

**Chapter Introduction**.
This chapter is rewritten and replaces the preprint of 9 October as well as my response to reviewer 1.

Lines 21 to 51, my interpretation: the research subject requires further introduction at the beginning, (too) much attention for statistical fitting as related to different values of the exponent in literature and without a focus on the goal of the manuscript. I propose to rewrite these sections with the note that Eq. 1 in the original manuscript will be removed and all equations are renumbered accordingly:

Competition among plants results in an allometric size-density relationship, where the average size of plants scales as a negative exponent of plant density. Theoretical mechanisms that explain the wide variation in empirically estimated values of the exponent are of interest because of the ubiquity of size-density relationships and practical significance for forestry and ecosystem management. Here, the size-density relationship is considered for the allometric relationship between average live aboveground biomass per tree in area $\bar{M}$ (g) and the number of trees in that area $N$ (m$^{-2}$), with the aim of developing a better explanatory model for the much-discussed variation in the exponent, after which the model is tested for interspecific density series of forests at broad spatial scale.

The allometric mass-density equation is traditionally written as (Yoda et al., 1963; Westoby, 1984):

$$\bar{M} = kN^{-\beta'} \tag{1}$$

where -β' is the scaling exponent and $k$ the scaling coefficient. This equation can also be written as:

$$\log\bar{M} = \log k - \beta'\log N \tag{2}$$

where -β' is the slope and $\log k$ the y-intercept of the regression line through datapoints in a log-log plot of average tree mass against tree density.

The exponent was some time thougt to converge to -3/2 in intraspecific as well as interspecific mass-density relationships, also referred to as the 'self-thinning rule' (Yoda et al., 1963; Gorham, 1979; Westoby 1984, White, 1985), until scrutiny showed that the slopes of the regression lines of time and density series vary much more (Weller 1987a; Li et al., 2005, 2006). Only for large interspecific datasets of plant communities at continental and global scales, where $N$ is the critical density of maximally packed individuals where all resources are used, an exponent value of -4/3 (or -3/4 as the exponent of $\bar{M}$, using $\bar{M}$ as the independent variable in Eq. 1) is reasonably well founded (Weller, 1989; Lonsdale, 1990; Deng et al., 2012; White et al., 2007). However, the generalisation of this value to mass-density relationships of tree-dominated communities of boreal, temperate, subtropical and tropical biomes separately seems to be unwarranted, as shown for the Chinese Forest Biomass Dataset (Luo, 1996, Li et al., 2006). Mechanistic model approaches based on geometric, allometric and dynamic growth arguments (e.g., Weller, 1987b; Adler, 1996; Enquist et al., 1998; Li et al., 2000; Deng et al., 2012) reproduce the general mass-density equation well, with plausible values of -β', but it remains problematic to predict the slopes of empirically obtained mass-density relationships (Reynolds and Ford, 2005).

Lines 52 to 72, my interpretation: These paragraphs discuss the light dependent normalization constant. The focus should be on the gradients in energy use in time series and density series of forests and not on the interpretation of light experiments. The light experiments will be interpreted in the Methods section. The paragraphs are replaced by the following text:

Here, an alternative approach is proposed to estimate the exponent value in forest mass-density relationship, with a focus on the intercept $\log k$, also referred to as a normalization constant that adjusts the general relationship across environments and species. The normalization constant is generally assumed to be a measure of whole stand resource use including light and reflects the assumption of a constant high rate of energy use in interspecific density series of crowded stands (Enquist et al., 1998; Deng et al., 2012). However, this assumption can be questioned in two ways. Firstly, it is commonly believed that the total leaf area of forest stands, measured as the leaf area index ($LAI$, m$^2$ m$^{-2}$), is constant when self-thinning occurs in dense stands, although $LAI$ varies with stand age and thus stand density (Ryan et al., 1997; Holdaway et al., 2008). This means that gradients in resource use can occur in time series of single stands and density series of spatially separated stands, as $LAI$ is a proxy for the resource use

of forests including light (Jump et al., 2017). Secondly, forest tree density increases with decreasing latitude to approximately 25°, while the total aboveground biomass is supposed to be constant (Enquist and Niklas, 2001) or increases, apart from the spatially restricted temperate rainforests (Pan et al., 2013). This means that the light dependent constant $k$, equal to $\overline{M}N^{\beta'}$ according to Eq. 1, increases with decreasing latitude, together with an increase in the sum of available solar energy use over the growing season $E_{sglob}$ (GJ m-2 yr-1). The consequence is that gradients in energy use can be expected in density series of forests at broad spatial scales.

After line 72: Inserted are the objective of the investigation (comment reviewer 1), as well as the hypothesis that is being evaluated and the reason that the hypothesis is relevant (reviewer 2):

The objective of this investigation is to develop an energy-biomass-density model (EBD) that uses a light absorption function including $LAI$ and $E_{sglob}$ instead of the normalization constant, enabling to account for gradients in energy use in density series of forests. Hamilton et al. (1995) reason that competition for light, rather than nutrients, causes mortality and an increase of leaf area shifts the limiting thinning line upwards. An increase of the nutrient supply increases the rate of progression of the thinning line, without changing the position of the line (Yoda et al., 1963; White and Harper, 1970). The evidence that nutrients also alter the position of the line (Westoby, 1984; Morris and Myerscough, 1991) is ambiguous. If nutrients change the position of the line, there may be an indirect effect on the leaf area and thus the capacity of plant stands to absorb light (Hamilton et al., 1995). This reasoning suggests that $LAI$ in the light absorption function links both energy and nutrient use to the forest mass-density relationship.

The hypothesis evaluated in this article is that the wide variation in the exponent in interspecific mass-density relationships can be reduced to the self-thinning rule in the EBD because the light absorption function accounts for possible gradients in $LAI$ and $E_{sglob}$ within density series of relatively undisturbed forests. The possibility that forest mass and density together strive for the same dynamic equilibrium with the regime of light absorption on broad spatial scales, which manifests in a single value of the exponent in the new model, would contribute considerably to a better understanding of the wide variation in empirically estimated mass-density relationships used in forestry (Vanderschaaf and Burkhart, 2007) and suitability of forest demographic models for Earth System Models (ESMs; Fisher and Serbin, 2017), leveraging the advancement in field measurements and remote sensing techniques to estimate $LAI$.

Line 73 to 92: These paragraphs are largely maintained, but the order is changed. The -1/2 self-thinning rule and the use of the dataset of Deng et al. (2012) are added. Two sentences have been added to highlight the relatively strong focus on statistical accountability compared to scientific explanations and interpretations. The paragraphs are replaced with the following text:

The hypothesis is evaluated for relatively undisturbed forests, because the incorporation of a light absorption function assumes a constant regime of light absorption over the years that is long enough to establish a dynamic equilibrium with the aboveground living biomass and tree density. Human, biotic and abiotic disturbances like thinning (not self-thinning), insect diseases and drought stress can lead to deviations in leaf area from this dynamic equilibrium, due to functional responses of forests to disturbances (Jump et al., 2017). The validity of the hypothesis is investigated by introducing the $LAI$ and $E_{sglob}$ separately and together in the light absorption function that replaces the normalization constant $k$. This results in three energy–biomass–density relationships or EBDs that

enable to examine how the interspecific mass–density relationship varies along gradients of leaf area and available solar energy separately and together. In addition to the three EBDs, the bivariate mass–density relationship and the relationship between leaf area and stand density are calculated, giving insight into how the slope of the mass–density line relates to a possible gradient in leaf area. The analysis focuses on the development of the exponent of $N$ because this is a central issue in the debate on the mathematical form of the self-thinning equation applied to forests. Additionally, we examine the extent to which the regression coefficients support the conclusions regarding the development of the exponent in the EBDs.

This article puts forward a thermodynamic interpretation of the EBD model, presenting it as a mathematical relationship between observations without the need for explanatory (sub)models of underlying processes (see Discussion). This results in a relatively stronger focus on statistical justification than on scientific explanations and interpretations. It has to be noticed that in this introduction is referred to the allometric mass–density relationship using $\overline{M}$, conform most cited studies, but the model development in this paper will be based on equations written in terms of the total living aboveground biomass $M$ (g m$^{-2}$) for statistical reasons, which means that the -3/2 self-thinning rule becomes a -1/2 self-thinning rule, while $k$ stays the same (see section 2.1). The dataset of relatively undisturbed forests to validate the hypothesis is selected from the compendium of Cannell (1982) that comprises standardized tabulations of field and experimental data of forests of approximately 600 reports worldwide (see 'Methods'). An indication of the representativeness of the selected stands is obtained by comparing the allometric mass-density relationship with a much larger data set of natural forests without $LAI$ data (Luo, 1996; Cannell, 1982) used by Deng et al. (2012).

**Chapter Methods**
**Section 2.1**, my interpretation:
More scientific explanation and interpretation are needed.

Lines 95 and 96, the first sentence is replaced by the text:
The introduction of a light absorption function begins with the formulation of a balance equation underlying the self-thinning equation. The self-thinning equation is used to describe the self-thinning process in time series of single plant populations. However, it is also applied to intraspecific and interspecific density series of spatially separated forests. This implicitly assumes a space-for-time approach, reflecting resource availability and functional differences among forest stands (Yu et al., 2024).

Line 114:

Insert after 'balance equation': (Eq. 4)

Lines 114 to 117, replace this sentence with the text:
Here, the exponent is expected to converge to the self-thinning rule, which shows in light experiments on monospecific even-aged plant populations, comparing self-thinning trajectories for different but constant levels of illumination. The exponent remains at a value of approximately -1/2 at each level of illumination, except for populations grown under deep shade. Meanwhile, the intercept of the thinning lines or normalization constant decreases as shade increases. This suggests that reduced light absorption due to a lower total leaf area, rather than increased shade, also results in a lower intercept of the thinning line, but only a different slope if light absorption

is not constant throughout the trajectory of self-thinning (Hiroi and Monsi, 1966; Lonsdale and Watkinson, 1982, 1983; Hutchings and Budd, 1981; Westoby and Howell, 1981; Westoby, 1984). The normalization constant, i.e. the integration constant in Eq. 2A, is not suitable to capture different levels of light absorption within time series or density series, which is adressed by the introduction of gradients in leaf area and the available solar radiation separately and together in the right-hand zero term of the balance equation (Eq. 4).

**Section 2.2**, my interpretation:
More scientific explanation and interpretation are needed.

Lines 123 to 125 are replaced by the text:
The LEBD is developed by introducing light capture in the balance equation, using the leaf area index or *LAI* of the forest stand (leaf area per unit of ground area in $m^2 m^{-2}$, one-sided for broadleaved trees and the projected leaf area for coniferous trees). Light availability within a canopy declines exponentially with increasing *LAI*, which implies that total light absorption is non-linearly related to total *LAI*. Light capture is calculated using *LAI* and the light extinction coefficient ε (dimensionless) in the adoption of Beer's Law (Monsi and Saeki, 1953):

Lines 147 and 148, this sentence is replaced by the text:

This means that only forests with *LAI* >1 will be included in the LEBD (Binckley et al., 2013). On the other hand, an upper limit for *LAI* may also be applicable because leaves in dim light environments can have very low assimilation rates close to their compensation points and therefore contribute little to the stand structure, as defined by the allometric mass-density relationship (Parker, 2020). An upper limit for LAI will be discussed in section 2.6. The validation of the LEBD against field data of forests results in values of the light extinction coefficient that are compared with literature values to determine whether the chosen mathematical approach and the data are appropriate to test the hypothesis (see Results).

**Section 2.3**, my interpretation:
More scientific explanation is needed, in addition to the response to reviewer 1.

Line 159, insert after 'latitude,':
altitude and hillshade,

**Section 2.4**, my interpretation: the scientific explanation is presented in the 'Introduction'. The scientific interpretation is presented in the 'Discussion'.

**Section 2.5**, my interpretation: The presentation of the fitting procedure for the different equations can be improved by using separate paragraphs.

Line 182: insert before starting a new paragraph in this line:
The equations are calculated using various statistical techniques.

Line 184: start a new paragraph in this line.

Line 186: start a new paragraph in this line.

Line 188: start a new paragraph in this line.

Line 192: start a new paragraph in this line.

Line 196: The reviewer comments that 'the climatological forcing comes from an obscure data source that I have never heard before.'
I suppose the source for climatological forcing data is unknown because it is primarily designed for companies. I have no indication that the data are insufficiently accurate for use in this study. I propose to add an explanatory text from the website on the first page of the Supplementary Material after https://www.soda-pro.com/web-services/meteo-data/monthly-means-solar-irradiance-temperature-relative-humidity:
The SoDa Service originates from a European project funded by the European Commission in 1999. A multi-disciplinary consortium has been assembled, which gathers companies and researchers with the necessary expertise in solar radiation and information and communications technologies. Customers and potential users are also represented as partners in the consortium via the involvement of commercial private vendors of solar radiation databases and of representatives of large international or local environmental research and development programs. The consortium:
   1. ARMINES (administrative co-ordinator) *France*
   2. MINES ParisTech - ARMINES - Centre Energétique et Procédés - Groupe Observation Modélisation et Décision (scientific co-ordinator) *France*
   3. MINES ParisTech - Les Presses MINES ParisTech *France*
   4. European Commission - Joint Research Centre *Italy*
   5. Meteorologiai Szolgalat - Meteorological Satellites Dpt *Hungary*
   6. METEOTEST *Switzerland*
   7. Carl von Ossietzky Universität Oldenburg Faculty of Physics *Germany*
   8. ICONS Srl *Italy*
   9. University of Manchester Institute of Science and Technology - Department of Physics *United Kingdom*
   10. Università degli Studi di Genova - Dipartimento di Fisica *Italy*
   11. CNRS - ENTPE - Département Génie Civil et Bâtiment *France*
   12. Fraunhofer - Institut fuer Solare Energiesysteme (ISE) *Germany*

**Section 2.6**, my interpretation: I suppose the comment 'The data sources are not well described' relates to this paragraph. The following adjustments are proposed:

Line 202: the following text is inserted before this line, with the second paragraph already suggested in response to reviewer 1:
The forest field and experimental data in Cannell's compendium (Cannell, 1982) include biomass data abstracted from about 600 publications (up to mid-1981), describing more than 1200 forest stands in 46 countries. The data are used in many biomass-density studies cited in this article (Weller, 1987a; Weller, 1989; Lonsdale, 1990; Enquist et al., 1998; Deng et al., 2012), with the difference that in this study the LAI data from Cannell's compendium are required for the LEBD and GEBD. The reliability of $LAI$ data varies due to, for instance, the number of sample trees in the experimental plots. The validity of the data, of which especially reliable estimates of $LAI$ are difficult to obtain (Bréda, 2010), can be assessed by the coefficient of determination in the model regressions and by comparing the extinction coefficient in the LEBD with literature values.
The data on $LAI$ of the selected stands were measured using litter traps, allometry based on sample trees or destructive methods, with the note that not all publications describe the measurement method. An example of the

use of litter traps is the deciduous woodland in Sweden (p. 222 Cannell, Supplementary Table 1), where the *LAI* value was averaged over a three-year measurement period. An example of the use of allometry are Pinus resinosa plantation forests in the U.S.A., New York (p. 314 Cannell, Supplementary Table 2), with 5 sample trees per plot in 4 plots and 3 sample trees in one plot. The *LAI* values were derived from regressions on breast-height diameter D. Examples of destructive methods are the Abies alba forest and deciduous forest in Czechoslovakia (p. 57 Cannell, Supplementary Table 1, 2). The *LAI* values were derived by multiplying the means of 5 sampled dominant trees, 5 co-dominants and 5 subdominants by the numbers of trees in each of these classes.

Line 209 and 210: the sentence beginning and ending in these two lines is replaced with the text:

Forests for which *LAI* was only estimated globally, for which different *LAI* values were estimated for the dry and wet seasons, or for which *LAI* was unclear due to a lack of distinction between trees and other plants, such as shrubs and undergrowth, were omitted. Stand data only obtained from published regressions elsewhere were also a reason for omission, especially as reliable estimates of *LAI* are difficult to obtain (Bréda, 2003). Only stands with an LAI > 1.0 were selected to stay within the validity limits set by the application of Beer's Law.

Line 211: insert after 'unclear':

(see section 2.2)

Line 218: the following text is added to the end of this line (this is also the response to reviewer 1):

Most of the data are from broadleaved forests, which can be considered a limitation. The latitude of the broadleaved forests varies between ca. 2° and ca. 56° N, with an average latitude of ca. 36° N. The latitude of the coniferous forests varies between ca. 32° and ca. 58° N, with an average latitude of ca. 41° N. The coniferous dataset includes thirteen forests with an altitude between 1500 and 2740 m.

**Section 2.7**: this section is already presented in response to reviewer 1. The second sentence in this response is replaced by the second and thirth sentence in the text hereafter:

**2.7 Notes on data from Deng et al. (2012)**

Dataset S1 from Deng et al. (2012) is used to obtain an indication of whether the selection of 199 forest stands with *LAI* data from Cannell (1982) is representative of a larger selection of forests for which *LAI* data are generally missing (see Discussion). The subset of natural forests in Deng's dataset S1 comprises 1109 forests from the Chinese Forest Biomass Dataset (Luo, 1996) and 241 forests from Cannell's (1982) compendium. Indicated are forest type, age, aboveground biomass, stand density, annual rainfall, annual mean temperature, potential evapotranspiration, altitude, latitude and longitude. The Luo database includes 6 forest biomes across the entire country. The 6 biomes are: boreal/alpine, temperate deciduous broadleaved, temperate coniferous, subtropical evergreen, subtropical coniferous and tropical rainforest/monsoon forest. Here, the 1350 forests are grouped into a boreal/alpine, temperate and subtropical/tropical group of forests, which are expected to differ in $E_{sglob}$ due to latitudinal and altitudinal gradients.

The interspecific forest mass-density relationship (Eq. 2A) is used to compare the slopes and intercepts of these three groups of forests with the data of the broadleaved and coniferous dataset of in total 199 forests including *LAI* data (see Discussion). The equation is also applied to all 1350 forests together. The boreal/alpine subgroup includes

252 coniferous and 13 broadleaved forests, the temperate subgroup 389 broadleaved and 219 coniferous forests and the (sub)tropical subgroup 335 broadleaved and 142 coniferous forests. The latitudes of the boreal/alpine subgroup (between ca. 27°-64° N) and the temperate subgroup (between ca. 27°-56° N) are about the same on average (ca. 40° N), but $E_{sglob}$ of the boreal/alpine subgroup is smaller on average due to the relatively short growing season of 166 alpine forests with a latitude between 27°-40° N and an altitude of ca. 2000 m or higher. So, the gradient in energy use is also related to an altitudinal gradient. The average latitude of (sub)tropical forests (between ca. 7°-38°) is about 28°. This includes forests in the southern hemisphere also.

**Chapter Results**

**Section 3.1:** The focus on the hypothesis is introduced in this section.

Lines 229 and 230, this sentence until 'is lower' is replaced by the text:

Based on the model results for the broadleaved dataset in Table 1, we can validate the hypothesis that the exponent in the EBDs converges to 0.50 when gradients in $LAI$ and $E_{sglob}$ are introduced stepwise in the light absorption function. In the interspecific mass-density relationship of Eq. 2A (Table 1, Fig. 1), the 0.22 exponent value

Line 246, insert after the period:

This is close to the hypothesised value of 0.50.

Lines 252 and 253, replace the sentence after 'and' with the text:

indicates that the mathematical design of the LEBD and stand data are aproppriate to correct for changes in the light capture of stands in the trajectory of decreasing stand density.

**Section 3.2:** The focus on the hypothesis is introduced in this section.

Lines 270, insert before this line the text:

The hypothesis that the exponent of $N$ converges to 0.50 in the EBDs is evaluated in the same way for both the coniferous and broadleaved datasets.

Line 281, insert after the period:

The exponent of 0.50 corresponds exactly to the hypothesised value.

Line 286 to 291, replace the text after 'Table 1' with:

corresponds with literature (Parker, 2020). The ε value decreases with increasing solar zenith angle (0 directly overhead) due to the generally planophile leaf canopies, compared to broadleaved forests with more random foliage orientation (Chen et al., 1997). The ε value is also lower because the light absorption capacity of the canopy is more affected by shoot clumping, i.e. leaves are more clumped on shoots compared to broadleaved forests (Kim et al., 2011). The close alignment of the extinction coefficient with values reported in the literature indicates that the mathematical design of the LEBD and stand data are appropriate to correct the density series for gradients in energy use.

**Chapter Discussion**

My interpretation: more scientific explanation and interpretation are needed, with a focus on the hypothesis of the investigation. The presented text below replaces my response to reviewer 1.

Line 328 to 336, these lines until the period in line 336 are replaced with the text:

To get an indication of whether the datasets of broadleaved and coniferous forests are representative, they are compared with each other as well as with the much larger data set of 1350 natural forests without $LAI$ data used by Deng et al. (2012).

Line 349 to 354, this paragraph is replaced with the text:

In the AEBD of Eq. 13 $\log(MN^{\alpha'})$ is expected to increase with increasing $E_{sglob}$ due to largely latitudinal gradients in global radiation. In the coniferous dataset, however, altitudinal gradients may also influence 13 stands between 1500 and 2740 m. The expected increase of $\log(MN^{\alpha'})$ is confirmed by the broadleaved and coniferous forests separately and together (Table 1) and is also visible in the intercepts of the mass–density relationships in Fig. 4, where the $\log k$ values increase in the order of boreal/alpine, temperate and (sub)tropical forests. The exponent values of α' = 0.56 and α' = 0.47 for the broadleaved and coniferous datasets, respectively, and α' = 0.52 for all stands together are not far from the self-thinning rule. The lower regression coefficient of f = 0.14 for the coniferous dataset compared to $f$ = 0.30 for the broadleaved dataset may be due to the lower competitive ability of many coniferous tree species at higher values of $E_{sglob}$. However, more data is needed to support this suggestion.

Line 371, add to this paragraph:

The GEBD calculates the scaling exponent correctly, but the regression coefficients in the GEBD are attenuated, due to measurement errors in the $LAI$ and $E_{sglob}$ (Aiken and West, 1991). This will be addressed in a forthcoming paper by analysing the structure of the underlying data, using intraspecific time and density series of forests with little or no biotic, abiotic and human disturbances.

Line 372 to 417, replace these paragraphs with the text that follows:

The hypothesis that the exponent in the EBD model converges to the self-thinning rule is not tested for intraspecific forest mass-density relationships, such as time series of single forest stands and intraspecific density series of neighbouring forests with a corresponding species composition. However, the self-thinning rule in its original interpretation as the upper boundary of mean plant biomass for a give plant density (Yoda et al., 1963; Osawa and Sugita, 1989) and the light experiments on monospecific even-aged plant stands cited before confirm the self-thinning rule for constant levels of light absorption at and below the upper boundary (see Sect. 2.1). Correcting deviations from the self-thinning rule due to light absorption gradients within time and density series using the light absorption function gives the exponent a completely different meaning.

The exponent -β in the general mass-density relationship (Eq. 2A) reflects the ratio of the relative growth rate and the relative mortality rate of the forest stand (Eq. 3). Variations in the exponent in time series of largely undisturbed forests have been shown to be a function of variations in environmental drivers, resource conditions (that might

also change with forest development), and forest properties (Yu et al., 2024). The variation in the value of the normalization constant among thinning lines has received relatively little attention (Dillon et al., 2019), but is correlated with the estimate of the exponent in the log - log relationship of Eq. 3 (Westoby, 1984). The normalization constant is described as a measure of a constant whole stand energy use (e.g., Westoby, 1984; White et al., 2007; Deng et al., 2012), with variations due to differences in resource use through time that are much stronger than variations in the exponent (Deng et al., 2006; Dai et al., 2009). This means that both the exponent and the normalization constant in the allometric mass-density relationship are a function of stand and environmental drivers, while in the EBDs the exponent is supposed to converge to a single value, the self-thinning rule, and only the light absorption function is a function of stand and environmental parameters.

This raises the question of how the introduction of a relatively simple light absorption function in the allometric mass-density relationship can reflect the myriad processes that affect the self-thinning process. A thermodynamic view on self-thinning can throw new light on this question, as thermodynamics can deal with the mathematical relationship between observations, without (sub)models of the underlying processes. The thermodynamic nature of the EBD model shows in natural forests, where individuals of all age classes are present simultaneously. Here, a steady state may ultimately be reached where the net production is equal to the losses, and changes in biomass tend to be zero. This means that the energy budget, as the sum of the ingoing and outgoing energy fluxes, is also zero, and it is not appropriate to specify the energy term $\log k$ in the mass-density relationship. However, from a thermodynamic point of view, entropy production is as important as energy. Following a further determination of the regression coefficients in the GEBD, the entropy production can be introduced into the GEBD, using the strong linear association with $E_{ssol}$ (e.g., Aoki, 1987, 1989; Brunsell et al., 2011), which allows the development of a physically correct thermodynamic equation. The thermodynamic equation describes the dynamic equilibrium of forest biomass and density together, as defined by the self-thinning rule, with the regime of entropy production. Time does not enter into the dynamic equilibrium described with the model, because the rate of biomass accumulation over time is not described (Westoby, 1984). The dynamic equilibrium can be seen as an attractor for forest succession after human, biotic and abiotic disturbances.

Deviations from the self-thinning rule may indicate disturbances that prevent the forest structure to be in a dynamic equilibrium with the use of energy and nutrients as formulated with the light absorption function. The use of EBDs and growth models together can provide further information on the interaction of changing environmental conditions with forest demography and carbon sequestration at different spatio-temporal scales, using the advancement in technology of field measurements and remote sensing techniques to estimate leaf area index and other stand indices like tree density and size (Yu et al., 2024). The better insight in $LAI$ and $E_{sglob}$ as predictors of stand structure and growth (see also Parker, 2020), together with high-resolution satellite or aerial remote sensing data at broad spatial scales, contributes to the inclusion of size-structured forest demographic models in Earth System Models (ESMs; Fisher and Serbin, 2017).

In forestry, the EBD model offers perspectives to further explain the variation in empirically determined self-thinning lines for different species/region combinations in relation to light (and nutrient) use, contributing to the reduction of errors in growth and yield models that are constrained by maximum size-density boundary lines (Vanderschaaf and Burkhart, 2007). Application of the EBD model to long-term trial plots is recommended, provided reliable $LAI$ data are available. Self-thinning lines of Norway spruce, European beech (Pretzsch et al., 2014) and Pinus sylvestris (Toraño Caicoya et al., 2024) do not show trends in time due to climate change-induced

longer growing seasons, only the growth rate increases. However, the intercepts of the self-thinning lines of Pinus sylvestris in Europe increase with decreasing latitude (Toraño Caicoya et al., 2024) and with increasing $E_{sglob}$, which is associated not only with a longer growing season but also with an increase in the intensity of solar radiation. This may lead to an increase in the intercept that is consistent with light experiments and the EBD model. Although the thermodynamic interpretation of the EBD model and the mechanistic models cited in this article represent different scales of inquiry and different scientific approaches, they can inform one another to obtain more insight into the common processes underlying these theories (Price et al., 2010).

New references compared to preprint of 9 October 2025:
Hamilton, N.S., Matthew, C. and Lemaire, G.: In defense of the -3/2 boundary rule: A re-evaluation of self-thinning concepts and status. Ann. Bot. 76, 6, 569–577, https://doi.org/10.1006/anbo.1995.1134, 1995.
Osawa, A., Sugita, S.: The self-thinning rule: Another interpretation of Weller's results. Ecology, 70, 279–283, https://doi.org/10.2307/1938435, 1989.
Parker, G.G.: Tamm review: Leaf Area Index (LAI) is both a determinant and a consequence of important processes in vegetation canopies. Forest Ecol. Manag. 477, 118496, http://dx.doi.org/10.1016/j.foreco.2020.118496, 2020.
VanderSchaaf, C.L. and Burkhart, H.E.: Comparison of methods to estimate Reineke's maximum size-density relationship species boundary slope. For. Sci., 53 (3), 435–442, https://doi.org/10.1093/forestscience/53.3.435, 2007.
Yu, K., Chen, H.Y.H., Gessler, A., Pugh, T.A.M., Searle, E.B., Allen, R.B., Pretzsch, H., Ciais, P., Phillips, O.L., Brienen, R.J.W., Chu, C., Xie, S., Ballantyne, E.P.: Forest demography and biomass accumulation rates are associated with transient mean tree size vs. density scaling relations. PNAS Nexus 3: 1–9, https://doi.org/10.1093/pnasnexus/pgae008, 2024.